# Metronidazole Potentiation by Panax Ginseng and *Symphytum officinale*: A New Strategy for *P. gingivalis* Infection Control

**DOI:** 10.3390/antibiotics12081288

**Published:** 2023-08-04

**Authors:** Salah M. Ibrahim, Abbas S. Al-Mizraqchi, Julfikar Haider

**Affiliations:** 1Department of Periodontics, College of Dentistry, University of Baghdad, Bab-Almoadham, Baghdad P.O. Box 1417, Iraq; 2Department of Basic Science, College of Dentistry, University of Baghdad, Bab-Almoadham, Baghdad P.O. Box 1417, Iraq; 3Department of Engineering, Manchester Metropolitan University, Manchester M1 5GD, UK; j.haider@mmu.ac.uk

**Keywords:** periodontal diseases, *Porphyromonas gingivalis*, *Symphytum officinale*, Panax Ginseng, metronidazole, biofilm inhibition, Acylated Homoserine Lactones

## Abstract

The important periodontal disease pathogen *Porphyromonas gingivalis* produces thick biofilms that increase its pathogenicity. Finding natural antimicrobial agents is crucial because of the rise in antibiotic resistance. The purpose of this study was to determine if plant extracts such as *Symphytum officinale* (S) and Panax Ginseng (G) were effective against *P. gingivalis* separately and in combination with a common antibiotic, metronidazole (F). Six different dilutions were produced using the plant extracts in different concentrations and antibiotics separately and in combination with F, G, and S using the two-fold serial dilution technique. To evaluate the effects of these substances, biofilm inhibition experiments were conducted. Plaque samples were collected from periodontitis patients to isolate *P. gingivalis*, and a standard strain of *P. gingivalis* (ATCC 33277) was purchased. Additionally, Acylated Homoserine Lactones (AHLs) detection was carried out to look for any activity that would interfere with quorum sensing. GraphPad Prism was used for statistical analysis with a *p*-value < 0.05. The combinations of *Symphytum officinale* and metronidazole (S+F) showed the maximum effectiveness in biofilm inhibition (98.7%), which was slightly better than G+F (98.2%), with substantial variations in biofilm inhibition levels in different treatment regimes. Notably, the patient isolate was more active than the standard strain. Additionally, the plant extracts and their combinations at particular dilutions had notable inhibitory effects on the generation of AHL (*p* < 0.05). The study highlights the possibility of *Symphytum officinale* and Panax Ginseng as effective treatments for *P. gingivalis* biofilm and AHLs, both on their own and in combination with metronidazole. These organic substances may open the door to cutting-edge methods of treating periodontal disorders.

## 1. Introduction

Globally, periodontal diseases pose a serious threat to public health because they affect a huge number of people worldwide and, if ignored, could result in tooth loss. One of the common oral bacteria present in the oral plaque of patients with periodontitis [1] and known to be actively involved in the progression of gingivitis is the Gram-negative anaerobic bacteria *Porphyromonas gingivalis* (*P. ginivalis*) [2]. On both soft and hard oral tissues, it is known to develop biofilms, an organized community of bacterial cells encased in a self-produced polymeric matrix [3]. These biofilms [4] are a characteristic of chronic periodontitis and provide resistance to host defense and common antimicrobial therapies, helping to enhance the pathogenicity of *P. gingivalis* [5].

A major concern for world health is antibiotic resistance, which has rendered many common antibiotics ineffective against bacterial infections. The search for new antimicrobial drugs, particularly those derived from natural sources, has been sparked by this dilemma [6]. Due to their effectiveness against a variety of bacterial infections, chemicals originating from plants have attracted enormous attention in this context [7,8]. Comfrey, also known as *Symphytum officinale*, and Panax Ginseng are two such promising herbs that have demonstrated exceptional antibacterial qualities in prior tests [1]. Furthermore, it is thought that the quorum sensing (QS) phenomenon in bacteria, a cell-to-cell communication activity, is crucial for biofilm formation and other virulence characteristics [9]. Acylated Homoserine Lactones (AHLs), signaling molecules found in many gram-negative bacteria, including *P. gingivalis*, are crucial for QS [10]. Therefore, preventing the synthesis of these molecules may be able to reduce bacterial pathogenicity and prevent the development of biofilms, making it a viable technique for treating chronic infections like periodontitis [6].

Allantoin, rosmarinic acid, and mucilage are just a few of the many bioactive substances found in *Symphytum officinale*, a plant that has long been used to cure a variety of diseases. The anti-inflammatory, antibacterial, and wound healing abilities of these substances have been studied in the past. An aphrodisiac, adaptogen, and all-purpose cure-all, Panax Ginseng is a well-known medicinal herb with East Asian origins [3]. The primary active ingredients in Panax Ginseng are saponins called ginsenosides, which have strong antibacterial and anti-inflammatory activities [11,12]. The investigation of these natural substances for their capacity to thwart quorum sensing and biofilm formation processes may eventually result in the development of novel approaches for periodontal disease therapy [10].

One of the main therapies for anaerobic bacterial infections, including those brought on by *P. gingivalis*, is the nitroimidazole antibiotic metronidazole. Metronidazole is very effective against anaerobic extracellular *P. gingivalis* by disrupting the DNA of anaerobic microbial cells [13]. The advent of antibiotic resistance and its associated negative consequences, however, have called into question its exclusive usage. Metronidazole may be more effective when combined with natural plant extracts, which would lower the dosage needed and the risk of adverse effects on human health [14]. Despite *Symphytum officinale* and Panax Ginseng having well-known antibacterial capabilities, nothing is known about how these two substances interact with *P. gingivalis*, specifically how they affect biofilm inhibition and AHL generation. Furthermore, few studies have examined the possible synergistic effects of using these extracts in conjunction with common antibiotics like metronidazole, though most of the investigations have concentrated on individual plant extracts [15,16,17]. By examining the effects of *Symphytum officinale* and Panax Ginseng on *P. gingivalis*, both separately and in combination with metronidazole, this study seeks to close this information gap. This study would make a substantial contribution to the creation of periodontal disease treatment plans [18,19].

A standard strain of *P. gingivalis* and a clinical isolate are utilized in this work in an effort to identify the extent of the effectiveness of the proposed treatment [20]. In light of this, the current study sought to assess the antibacterial effectiveness of Panax Ginseng and *Symphytum officinale* extracts against *P. gingivalis*. The experiment was then expanded to look at how these extracts affected the growth of AHL and *P. gingivalis* biofilm when combined with metronidazole, a medication frequently used to treat periodontitis. The effects of a common strain of *P. gingivalis* and a patient-derived clinical isolate were compared, providing information on the possible therapeutic uses of these plant extracts.

## 2. Materials and Methods

### 2.1. Bacterial Strains and Cultures

The main bacterial strain used in this investigation was the well-known laboratory strain *P. gingivalis* ATCC 33277 (American Type Culture Collection). Additionally, a clinical isolate of *P. gingivalis* was included in this study to assess the application of the research findings. The plaque samples were collected from periodontitis patients. Informed consent was obtained from all subjects involved in the study (Ethical approval Reference number 382, University of Baghdad, Iraq). The injection of Gracey-curette enabled the collection of subgingival plaque specimens. When the curette touched the bottom of the periodontal pocket without damaging the soft tissues, it collected subgingival plaque. Following that, the plaque sample was immersed in sodium thioglycolate.

Both the standard strain and clinical isolate of *P. gingivalis* were grown anaerobically in brain heart infusion (BHI) [21] broth that had been treated with hemin and vitamin K while maintaining a temperature of 37 °C.

### 2.2. Plant Extracts and Antibiotic Working Concentrations

Both Panax Ginseng and *Symphytum officinale* (comfrey) were purchased from reputable vendors (Rejuvica Health, Gilbert, AZ, USA). Aqueous extracts of Panax Ginseng and comfrey were sterilized using millipore disposable filters (0.22 µm). The selected antibiotic, metronidazole, was purchased from a local pharmacy. Dimethyl sulfoxide (DMSO) was used to dissolve all substances in order to create workable dilutions from the base concentrations. Experimental extracts include each of *Symphytum officinale* and metronidazole (S+F), followed by *Symphytum officinale* (S) alone, then Panax Ginseng and metronidazole (G+F), then Panax Ginseng (G) alone, and finally metronidazole (F) alone. Working concentrations for metronidazole, Panax Ginseng, and *Symphytum officinale* were produced both alone and in combination. To enable a thorough investigation of each chemical’s effects throughout a variety of doses, six dilutions (D1 to D6) were generated for each compound using a systematic two-fold serial dilution technique, as shown in Table 1. Plant extracts and metronidazole (base concentrations: S = 330 mg/mL, G = 1000 mg/mL, and F = 500 mg/mL) without any dilution were considered the controls. Total combination concentration (c3) was determined by using Equation (1).
v1 × c1 + v2 × c2 = v3 × c3(1)
where v1/v2 is 1.0 mL as the volume of individual elements (G, F, or S), c1/c2 is the base concentration of individual elements, and v3 is a combination volume of 5 mL. For example, in order to calculate the combination concentration of S+F, volume of S or F, v1 = v2 = 1 mL, base concentration of S, c1 = 330 mg/mL, base concentration of F, c2 = 500 mg/mL, and combination volume v3 = 5 mL were considered, and following Equation 1, the combination concentration c3 is calculated as 166 mg/mL. According to the two-fold serial dilution, half of this concentration is 83 mg/mL, which is presented in the second row and last column.

### 2.3. Biofilm Inhibition Assay

*P. gingivalis* was grown in a brain-heart infusion medium containing 10% blood and 0.2 mg/mL vitamin K. (Sigma-Aldrich, St. Louis, MO, USA). Bacteria thrived in an anaerobic chamber with 5% H_2_, 10% CO_2_, and 85% N_2_ at 37 °C for 48 to 72 h [22]. The diameters of the inhibition zones formed around the extracts were measured in millimeters with a metric ruler [23].

Using a microtiter plate biofilm test, the potential of plant extracts and metronidazole to prevent *P. gingivalis* from forming biofilms was evaluated. Both the standard strain and the isolate were exposed to various test drug doses. The microplates were then incubated anaerobically at 37 °C for 24 h, and the amount of biofilm inhibition was measured using crystal violet staining [24]. The variations in optical densities (ODs) were used to quantify the percentage of biofilm inhibition. The isolates were deemed strong if the OD was greater than 0.240, moderate if the OD was between 0.120 and 0.240, and weak if the OD was less than 0.120.

### 2.4. Detection of Acylated Homoserine Lactones (AHLs)

AHLs, a crucial component of *P. gingivalis*’s quorum sensing, were determined using a colorimetric technique [25]. At different dilutions, the effects of *Symphytum officinale*, Panax Ginseng, and metronidazole on the generation of AHLs by two *P. gingivalis* strains were observed in Figure 1.

The potential of the plant extracts to inhibit quorum sensing was assessed by using a colorimetric method, which produces orange pigmentation in the presence of AHL, a quorum sensing molecule. The reported strains were co-cultured with *P. gingivalis* in the presence and absence of plant extracts. The intensity of orange pigmentation was taken as a measure of AHL activity and quantified spectrophotometrically. The isolates were deemed to produce weak or negative AHLs (negative control) if the OD was less than 0.98 and strong or positive AHLs (positive control) if the OD was greater than 0.98 [26].

### 2.5. Statistical Analysis

All statistical analyses were conducted with the assistance of the Statistical Package for Social Sciences (SPSS) version 24.0. All experiments were performed in triplicate, and the data were expressed as mean ± standard deviation. Statistical analyses were performed using one-way ANOVA, followed by Tukey’s multiple comparison tests. *p*-values < 0.05 were considered statistically significant.

## 3. Results

### 3.1. Antibacterial Activity of Plant Extracts and Metronidazole

The initial assessment was based on the antibacterial properties of metronidazole, Panax Ginseng, and *Symphytum officinale* (Comfrey). The antibacterial efficacy of the plant extracts and metronidazole against the standard bacterial strain *(P. gingivalis* ATCC 33277) and the clinical isolate was evaluated separately and in combination. The outcomes showed clear antibacterial activity, with diverse effects at various doses (Table 1). Only D4 was nonsignificant for the strains (*p* > 0.05) (Table 2). While the lowest concentration of D5 did not show any antibacterial activity at all. Therefore, no further testing was carried out with D6. The experiments were carried out three times, and the inhibition-zone diameters are presented in graphical form in Figure 2.

### 3.2. Biofilm Inhibition Rate of Bacterial Standard Strain and Isolates

Under the influence of several experimental extracts, the biofilm inhibition rates of the standard and isolated bacterial strains were compared. The substantial antibacterial activity of the test compounds was shown by the considerable variation in biofilm inhibition across the various treatments (*p* < 0.05) (Table 3). It should be noted that even the lowest dilution of D6 produced some antibacterial effects.

### 3.3. Acylated Homoserine Lactones (AHLs) Production

Finally, it was assessed how the plant extracts and the antibiotic metronidazole affected the creation of AHLs, an essential component of bacterial communication. As indicated in Table 4, the plant extracts and metronidazole significantly inhibited the formation of AHLs (*p* < 0.05). The experiments were carried out three times.

Overall, the findings strongly suggested that metronidazole, Panax Ginseng, and *Symphytum officinale* (Comfrey) have antibacterial and biofilm inhibitory effects, suggesting that they may be used to treat *P. gingivalis* infections. The fact was that these results remained true for both the common laboratory strain and the clinical isolate, indicating their significance in the actual clinical environment (Figure 3).

### 3.4. Dose-Dependent Effects of G, S, and F

The percentage inhibition of biofilm formation was calculated from optical density data using Equation (2) in order to determine the effect of doses on the biofilm.
(2)$$\%\;\mathrm{biofilm}\;\mathrm{inhibition}\;=1-\frac{\mathrm{OD}\;\mathrm{with}\;\mathrm{treated}\;\mathrm{extract}\;}{\mathrm{OD}\;\mathrm{with}\;\mathrm{nontreated}\;\mathrm{control}}\times 100$$

OD with treated extract data for different doses was taken from Table 3, and OD with a nontreated control was calculated by averaging the OD values of the controls for both the standard strain (2.846) and clinical isolate (3.186) in Table 3.

The results demonstrated the dose-dependent nature of biofilm inhibiting effects with different doses of G, S, and F (Table 1). The D1 dose produced the most effective results with respect to the values of the extract combination. However, S+F produced the maximum inhibition percentage in the case of the standard strain, whereas G+F produced the maximum inhibition in the case of the isolated strain. The results of this study will be extremely useful in determining the combinations and concentrations of each medication that will have the greatest therapeutic impact (*p* = 0.00014) (Table 5).

### 3.5. Correlation between Biofilm Inhibition and AHL Production

Figure 4 presents the relationship, if any, between biofilm inhibition represented by optical density and AHL activity for the clinical isolate and standard strain for all five cases (G, S, F, G+F, and G+S). In all cases, some degree of linear correlation existed, and this was associated with the correlation coefficient (R^2^) values presented within the graphs. This significant association could imply that the biofilm inhibition and anti-biofilm actions of these medications are, at least in part, influenced by the disruption of bacterial quorum sensing mechanisms. For both the standard and isolated strains, the combination of Panax Ginseng and metronidazole (G+F) produced the best correlation for both stains.

## 4. Discussion

The current investigation focused on both a standard strain and clinically isolated bacteria to examine the inhibitory potential of Panax Ginseng (G), *Symphytum officinale* (S), and metronidazole (F) against *P. gingivalis*. When exposed to the tested treatments, a substantial suppression of biofilm formation was demonstrated by both the standard strain and the clinical isolate. The results support earlier research showing that *P. gingivalis* is very amenable to biofilm reduction through a variety of treatment approaches, both conventional and modern [27], and that G, S, and F have powerful antibacterial and anti-biofilm capabilities [3].

Additionally, it appeared that the combination therapies (G+F, S+F) had a more prominent impact than the separate treatments, which may have been the result of a synergistic action by the two [14]. The generation of AHL, a crucial component of bacterial quorum sensing, was successfully reduced by the tested treatments in both the clinical isolate and the standard strain, according to the research. The capacity of *P. gingivalis* to coordinate group behavior may be compromised by this restriction of AHL synthesis, which would reduce the pathogenic potential of the organism [10].

A possible strategy for reducing bacterial pathogenicity is to disrupt quorum sensing processes. It is possible to significantly lessen the pathogenic potential of bacteria like *P. gingivalis* by interfering with their coordination and communication. Bacterial quorum sensing, which is mediated by signal molecules like AHLs, has become a fascinating topic of research, especially in connection to bacterial pathogenicity and the development of durable biofilms [6]. Notably, the current findings confirm the biofilm-inhibitory effects found by providing strong evidence that G, S, and F can impair quorum sensing with *P. gingivalis* by preventing the generation of AHL. This capacity to interfere with the quorum sensing systems is consistent with other investigations that have noted the inhibitory effects of natural extracts on quorum sensing [28].

Intriguingly, the bacteria isolated from patients were more energetic and active than the conventional strain, as evidenced by the smaller inhibition zone diameter (Table 2), higher optical density (Table 3), and higher AHL activity (Table 4). This also corroborates other studies that claim clinical isolates may display unique characteristics [29]. The complexity of bacterial behavior in actual clinical settings is highlighted by the increased activity that was seen in clinical isolates compared to the reference strain. It serves as a reminder that bacterial pathogenicity and medication resistance can differ greatly amongst strains, depending on a number of variables, including their genetic make-up and the environment from which they are isolated [30]. However, the specific causes of this disparity are yet unknown and need more research.

Depending on the treatment concentration, a sizable variance in the observed outcomes was noticed, pointing to a strong dose-dependent response for the inhibitory effects of G, S, and F. This is in line with other studies showing that the concentration of antibacterial drugs frequently affects their effectiveness [31]. Other research investigating the antibacterial activity of natural extracts has shown that such a reaction is a typical characteristic of many antimicrobial compounds [1]. These dose-response correlations are essential for pinpointing the precise concentrations that can produce the best therapeutic outcomes while minimizing any side effects. When considered as a whole, these results highlight the promising antibacterial and anti-biofilm activities of G, S, and F against *P. gingivalis*, suggesting the potential for innovative treatments to avert or cure infections caused by bacteria that form biofilms.

This work adds new knowledge to the expanding body of research that supports the use of natural extracts as alternatives or supplements to conventional antimicrobial treatments by confirming the ability of G, S, and F to disrupt these bacterial communication routes [31]. Additionally, the synergistic effects of combining therapies (G+F, S+F) provide new avenues for creating more powerful and comprehensive treatment plans. The combination of several antibacterial drugs frequently leads to increased antibacterial effectiveness, as shown by the findings and those of other researchers [32]. Therefore, it is important to investigate the ramifications of these synergistic effects, including any potential effects on lowering the probability of bacterial resistance.

The results of this investigation shed light on the efficacy of metronidazole (F), *Symphytum officinale* (S), and Panax Ginseng (G) as antibacterial and anti-biofilm agents against *P. gingivalis*. Further in vivo research is needed to verify these findings. The therapeutic potential and safety profile of these medicines might be further understood using animal models or human participants in clinical studies. While it was found in this study that G, S, and F can hinder quorum sensing in *P. gingivalis* by blocking the production of AHL, the fundamental mechanisms underpinning this process are still not fully understood. The particular molecular processes by which these natural extracts and antibiotics disrupt quorum sensing pathways need further investigation. This information will aid in the creation of more focused and successful therapy approaches.

This research is limited to providing a large number of samples for microbiological analysis using advanced techniques.

## 5. Conclusions

In conclusion, this study focuses on examining the inhibitory potential of Panax Ginseng (G), *Symphytum officinale* (S), and metronidazole (F) against *P. gingivalis* collected from a standard strain and clinical isolate. The findings confirmed that *P. gingivalis* was susceptible to several biofilm-reduction therapy modalities. The antibacterial and anti-biofilm properties of G, S, and F, the tested treatments, showed a significant inhibition of biofilm formation against both strains. Combination therapy (G+F or S+F) had a more pronounced effect than the individual therapies, possibly as a result of their synergistic activities.

## Figures and Tables

**Figure 1 antibiotics-12-01288-f001:**
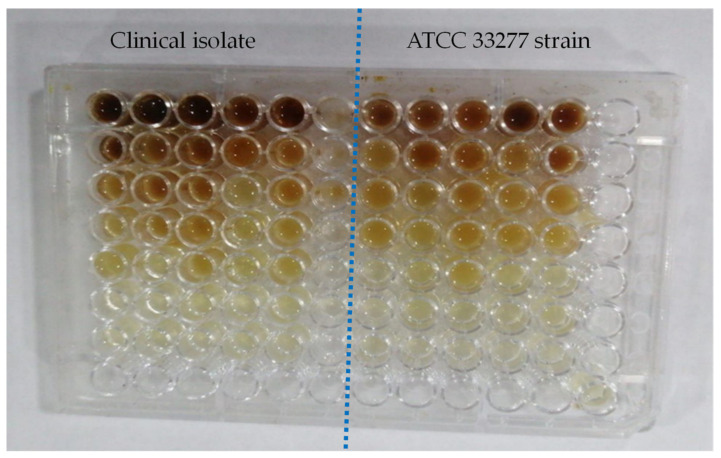
Detection of Acylated Homoserine Lactones (AHLs) for *P. gingivalis* (standard strain and isolated bacteria).

**Figure 2 antibiotics-12-01288-f002:**
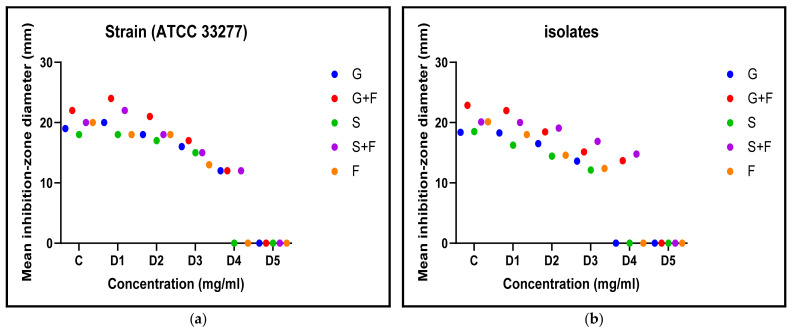
Antibacterial inhibition zone diameters created by plant extracts and metronidazole at different doses against *P. gingivalis* (**a**) standard strain and (**b**) clinical isolate.

**Figure 3 antibiotics-12-01288-f003:**
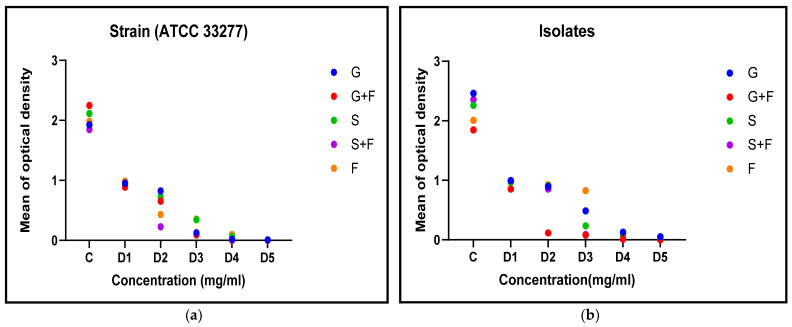
Acylated Homoserine Lactones (AHLs) activity at different doses for *P. gingivalis* (**a**) standard strain and (**b**) clinical isolate.

**Figure 4 antibiotics-12-01288-f004:**
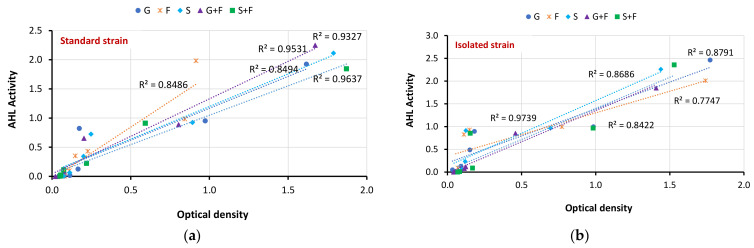
Relationship between biofilm inhibition and AHL activity at different doses for *P. gingivalis* (**a**) standard strain and (**b**) clinical isolate.

**Table 1 antibiotics-12-01288-t001:** Total concentrations of plant extracts and metronidazole.

Dilution	Total Concentrations (mg/mL)				
	G	S	F	G+F	S+F
C: Control	1000	330	500	300	166
D1: first dilution	500	165	250	150	83
D2: second dilution	250	82.5	125	75	41.5
D3: third dilution	125	41.25	62.5	37.5	20.75
D4: fourth dilution	62.5	20.625	31.25	18.75	10.375
D5: fifth dilution	31.25	10.312	15.625	9.375	5.187
D6: sixth dilution	15.625	5.156	7.812	4.687	2.593

**Table 2 antibiotics-12-01288-t002:** Antibacterial activities of plant extracts and metronidazole.

Microorganisms	Concentration	Mean Inhibition-Zone Diameter, ± SD (mm)					F-Test	*p*-Value
		G	F	S	G+F	S+F		
Strain (ATCC 33277)	C	20.5120 ± 0.014	20.4119 ± 0.114	18.6111 ± 0.147	24.0125 ± 0.321	22.0367 ± 0.014	132.25	0.000
	D1	20.2476 ± 0.214	18.2778 ± 0.0251	18.2110 ± 0.202	22.5019 ± 0.052	20.8970 ± 0.023	129.63	0.005
	D2	18.3017 ± 0.105	18.2100 ± 0.221	17.1556 ± 0.054	21.000 ± 0.014	18.0000 ± 0.514	141.11	0.003
	D3	16.1506 ± 0.221	13.6321 ± 0.074	15.0556 ± 0.015	17.8642 ± 0.0162	15.6410 ± 0.0363	128.63	0.001
	D4	12.000 ± 0.0241	Zero	Zero	12.2590 ± 0.0669	12.3780 ± 0.0554	99.52	0.142
	D5	Zero	Zero	Zero	Zero	Zero	-	-
Isolate	C	18.3889 ± 0.041	20.1359 ± 0.014	18.5124 ± 0.025	22.8690 ± 0.0021	20.1056 ± 0.0145	138.28	0.0025
	D1	18.2778 ± 0.25	18.0189 ± 0.0036	16.2531 ± 0.021	21.9810 ± 0.084	20.0245 ± 0.0126	129.69	0.0015
	D2	16.5000 ± 0.019	14.5890 ± 0.033	14.4376 ± 0.033	18.4587 ± 0.0415	19.0849 ± 0.0235	137.18	0.0032
	D3	13.6111 ± 0.014	12.3801 ± 0.74	12.1400 ± 0.039	15.1398 ± 0.0963	16.8941 ± 0.0229	98.69	0.0015
	D4	Zero	Zero	Zero	13.6782 ± 0.254	14.7901 ± 0.0325	91.57	0.521
	D5	Zero	Zero	Zero	Zero	Zero	-	-

Note: C is the control group concentration; D1 to D5 are 2-fold serial dilutions (Table 1); Zero means no inhibition zone; Panax Ginseng (G) and *Symphytum officinale* (S); metronidazole (F); Panax Ginseng + metronidazole (G+F); *Symphytum officinale* + metronidazole (S+F).

**Table 3 antibiotics-12-01288-t003:** Mean optical density (OD) of biofilm formed by *P. gingivalis (strain ATCC 33277)* and isolate.

Microorganism	Concentration	Mean Optical Density (OD) of Biofilm ±SD					F-Test	*p*-Value
		G	F	S	G+F	S+F		
Standard strain (ATCC 33277)	C	2.846	2.846	2.846	2.846	2.846	124.69	0.0041
	D1	1.616 ± 0.071	0.914 ± 0.14	1.788 ± 0.002	1.672 ± 0.302	1.870 ± 0.078	125.23	0.0062
	D2	0.973 ± 0.062	0.843 ± 0.092	0.892 ± 0.056	0.804 ± 0.041	0.592 ± 0.041	133.36	0.0024
	D3	0.173 ± 0.035	0.227 ± 0.047	0.247 ± 0.105	0.203 ± 0.004	0.219 ± 0.009	122.56	0.0093
	D4	0.165 ± 0.030	0.147 ± 0.081	0.200 ± 0.010	0.077 ± 0.092	0.074 ± 0.106	136.63	0.004
	D5	0.113 ± 0.025	0.109 ± 0.037	0.112 ± 0.053	0.035 ± 0.105	0.055 ± 0.003	115.96	0.0082
	D6	0.077 ± 0.008	0.079 ± 0.069	0.073 ± 0.006	0.021 ± 0.038	0.050 ± 0.014	121.52	0.0093
Clinical isolate	C	3.186	3.186	3.186	3.186	3.186	119.52	0.004
	D1	1.772 ± 0.080	1.530 ± 0.079	1.440 ± 0.250	1.741 ± 0.002	1.408 ± 0.004	126.39	0.005
	D2	0.985 ± 0.068	0.983 ± 0.130	0.695 ± 0.091	0.772 ± 0.031	0.459 ± 0.045	127.25	0.000
	D3	0.182 ± 0.034	0.154 ± 0.056	0.125 ± 0.003	0.149 ± 0.003	0.124 ± 0.290	132.78	0.0063
	D4	0.151 ± 0.045	0.169 ± 0.085	0.120 ± 0.018	0.111 ± 0.092	0.114 ± 0.007	133.03	0.0025
	D5	0.091 ± 0.028	0.082 ± 0.101	0.098 ± 0.004	0.070 ± 0.013	0.052 ± 0.098	132.96	0.0014
	D6	0.034 ± 0.013	0.069 ± 0.040	0.045 ± 0.013	0.056 ± 0.032	0.041 ± 0.002	128.55	0.0069

Note: C is the control group concentration; D1 to D6 are 2-fold serial dilutions (Table 1); Panax Ginseng (G) and *Symphytum officinale* (S); metronidazole (F); Panax Ginseng + metronidazole (G+F); *Symphytum officinale* + metronidazole (S+F).

**Table 4 antibiotics-12-01288-t004:** Influence of plant extracts and metronidazole on AHL production.

Microorganism	Concentration	AHL Activity ± SD					F-Test	*p*-Value
		G	F	S	G+F	S+F		
Standard strain (ATCC 33277)	C	1.926 ± 0.379	1.984 ± 0.630	2.114 ± 0.049	2.248 ± 0.801	1.845 ± 0.778	88.95	0.0014
	D1	0.953 ± 0.078	0.987 ± 0.122	0.925 ± 0.500	0.887 ± 0.201	0.914 ± 0.058	41.24	0.0036
	D2	0.824 ± 0.135	0.432 ± 0.004	0.725 ± 0.001	0.654 ± 0.015	0.226 ± 0.031	51.14	0.000
	D3	0.127 ± 0.039	0.354 ± 0.062	0.344 ± 0.023	0.092 ± 0.002	0.114 ± 0.106	44.56	0.0025
	D4	0.017 ± 0.025	0.098 ± 0.100	0.065 ± 0.013	0.001 ± 0.005	0.023 ± 0.010	39.23	0.0051
	D5	0.009 ± 0.001	0.006 ± 0.004	0.006 ± 0.003	0.000	0.005 ± 0.001	33.54	0.0052
Clinical isolate	C	2.462 ± 0.980	2.009 ± 0.001	2.259 ± 0.979	1.845 ± 0.002	2.357 ± 0.368	29.68	0.0041
	D1	0.997 ± 0.060	0.995 ± 0.029	0.965 ± 0.019	0.854 ± 0.031	0.967 ± 0.002	36.05	0.002
	D2	0.895 ± 0.143	0.927 ± 0.101	0.912 ± 0.103	0.115 ± 0.003	0.854 ± 0.251	41.98	0.0042
	D3	0.487 ± 0.009	0.827 ± 0.058	0.235 ± 0.072	0.078 ± 0.010	0.089 ± 0.050	42.56	0.003
	D4	0.129 ± 0.020	0.094 ± 0.010	0.059 ± 0.002	0.012 ± 0.020	0.012 ± 0.021	40.25	0.000
	D5	0.05 ± 0.003	0.009 ± 0.013	0.001 ± 0.010	0.000	0.002 ± 0.001	40.14	0.009

Note: C is the control group concentration; D1 to D5 are 2-fold serial dilutions (Table 1); Panax Ginseng (G) and *Symphytum officinale* (S); metronidazole (F); Panax Ginseng + metronidazole (G+F); *Symphytum officinale* + metronidazole (S+F).

**Table 5 antibiotics-12-01288-t005:** Percentage inhibition of biofilm formation for bacterial strains used.

Microorganism	Concentration	Percentage Inhibition of Biofilm Formation				
		G	F	S	G+F	S+F
Standard strain (ATCC 33277)	D1	98.1	97.8	98	98.2	98.7
	D2	96.1	97.4	96.9	97.8	98.3
	D3	95.2	94.6	95.2	96.5	96.4
	D4	94.2	95.1	94	95.3	96.1
	D5	69	69.1	78.1	80.7	85.5
	D6	44.3	51.9	54.8	45.3	55.8
Clinical isolate	D1	97.2	97.2	97.4	99.2	98.2
	D2	96	96.1	96.1	98.7	98
	D3	94.2	94.8	92.9	97.2	97.3
	D4	93.9	92	91.3	92.8	92.1
	D5	69.9	70.3	68.6	71.7	79.1
	D6	43.2	67.8	37.1	41.2	34.2
X^2^	3.254					
*p*-value	0.00014					

## Data Availability

Upon reasonable request, the data created and examined during the current study can be made available.

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
