# Peer review of "Metronidazole Potentiation by Panax Ginseng and Symphytum officinale: A New Strategy for P. gingivalis Infection Control"

_antibiotics, 2023, doi:10.3390/antibiotics12081288_

Round 1

Reviewer 1 Report

Metronidazole Potentiation by Panax Ginseng and Symphidium Officinale: A New Strategy for P. gingivalis Infection Control

by Ibrahim et al.

In this paper, the authors attempt to show synergistic activity between Panax Ginseng and Symphidium Officinale. The manuscript is poorly-written, and is not proofread properly. There is too much emphasis being place on zone of inhibition, which is a good qualitative method but is not meant to be quantitative. Results have not been confirmed by doing any of the studies in broth.  There are unnecessary figures showing the same results as in the tables. Figure 1 probably has the same petri dish photographed twice. The last figure shows error bars greater than the averages suggesting that the numbers are not significant or reliable. A positive aspect of the paper is that there is probably some synergy in the activities of G, S and F.

Some detailed comments on the paper are as follows:

Line 13: “produce large biofilms” Not clear what “large” means.

Line 17: “were produced the plant extracts” Sentence incomplete: some word is missing

Line 18: “combination with G and F and S and F” Wrong use of the word “and”. Suggestion: “combinations of F with G and S”.

Line 19: Change “substances biofilm” to “substances, biofilm”

Line 24: “(98.7%), which was better than G+F (98.2%)” 98.7% can hardly be said to be better than 98.2%. They are about the same.

Line 26: “the isolate…..the patient” There were multiple “patients”, as mentioned in an earlier sentence.  So which specific “the isolate” and “the patient” are being discussed here?

Line 48: “sparked by this dilemma” No dilemma has been mentioned.

Line 70: “It causes harm to DNA” Please be more specific. What is meant by “harm to DNA”? Actually it works by degrading DNA. Please provide a reference.

Line 71: “prevents bacteria from making proteins” If DNA is degraded, obviously, protein will not be made eventually. That does not justify saying that the antibiotic prevents protein synthesis. Is there a recent reference for this? In the old days, before the DNA cleavage mechanism was discovered, many other mechanisms of action were erroneously assumed for this antibiotic.

Line 105: Change “culture condition” to “cultures”

Line 105: Figure 1 a and b. It may sometimes happen that two streak patterns may appear to be similar if the researcher has a consistent way of streaking. However, in this case the two streak patterns appear to be EXACTLY the same. It may be possible that the same plate has been photographed twice. Also, the streaking does not follow standard microbiological technique.

Line 105: The purpose of Figure 1 is not clear. The reader gets no new information from this. Pictures of petri dishes (especially poorly streaked ones) are not published in science journals unless any new information is presented. If the purpose is to show the colony phenotypes, then show enlarged pictures of single colony from each. Picture (a) is blurry and of different color than (b). My suggestion is to delete the figure because it is too trivial.

Line 108: “A suitable solvent liquid herbal extract” The language makes no sense.

Line 109: Change “addative” to “additive”

Line 109: “20” This whole sentence is not clear.  It is also not clear what the reference is for.

Line 112: “workable concentrations” Mention the concentrations of each.

Line 114: Change “Ginsengand and” to “Ginseng and”

Line 114: Add space before “+”

Line 115: Add space before “(G)”

Line 120 Table 1: Please confirm that the mentioned concentrations of G+F and S+F are total concentrations. If not, please write “each” in the heading (if they are in equal amounts).

Line 120 Table 1: What is the basis of the concentrations of G+F and S+F? I see that S+F numbers are half of the S numbers. How were the G+F numbers decided? If each component in the mixtures were in equal amount, why were they taken in equal amounts and in proportion to the concentrations that were used alone?

Line 124: Why is 5% H2 needed? Is it really H2? Any safety precautions taken?

Line 124: I checked the reference (21) provided but this information is not there in the paper.

Line 131: Change “Moderate” to “moderate”

Line 139: “Figure Error! No text of specified style in document.2.” Not clear what this means. I guess, it should be “Figure 2.”  This is again repeated in line 165.

Line 133 and 141: Not clear why Sections 2.4 and 2.5 are two separate sections.

Line 147-149: “the negative…..molecules” Change “the” to “The”. This sentence is lacking a verb.

Line 166 Table 2 and many other places: “inhibition zone” It is more commonly called “zone of inhibition”. If the authors insist on writing the former, the two words should be joined by a hyphen (inhibition-zone).

Line 166 Table 2: For columns 3-7, the heading says, “Mean”.  If these numbers are the means, a better way of presenting the numbers is to also include the standard deviations along with the means (X±Y). Also, please mention how many times the experiment was done in each case.

Line 166 Table 2: All abbreviations should be explained in the table footnote or referred to the right place (e.g. for D1 – D5 see Table 1). The abbreviation, D is used for both diameter and concentration, resulting in confusion. What is C? Is it a control? Please mention the concentrations of the antibiotics in C. Are we supposed to compare the D’s with the C’s? If you don’t mention what C is, I am not sure what this table is about.

Line 166 Table 2: The method for zone of inhibition determination has not been described anywhere. It should be described in the Methods section. I assume that the antibiotics were soaked onto filter paper disks and then placed on plates. Do these numbers in Table 2 include the diameter of the antibiotic containing disk? If that is so, what is meant by zero mm diameter reported for D4 and D5? If all the numbers include the diameter of the disks, then no number can be zero.

Line 166 Table 2: I think, for better comparisons, the order of the columns should be G, F, S, G+F, S+F.

Line 170 Figure 2: The purpose of this figure is not clear. It presents the same data as in Table 2. The figure does not provide any new information and actually adds to the confusion.

Line 177 Table 2: This should be Table 3.

Line 177 Table 3: What are the numbers in this table? Whatever these numbers represent, that should be the heading in row 1, instead of “Drugs and Extracts”

Line 177 Table 3: This table makes absolutely no sense to me. Since you are measuring biofilm by optical density readings, why not simply present the OD values?

Line 183 and 185: “Table 3” Should be “Table 4”

Line 185 Table 4: Once again it is not clear what C is. Why do Table 2 and 4 have C but Table 3 does not?

Line 185 Table 4: I think, for better comparisons, the order of the columns should be G, F, S, G+F, S+F.

Line 185 Table 4: For columns 3-7, the heading says, “Mean”.  If these numbers are the means, a better way of presenting the numbers is to also include the standard deviations along with the means (X±Y). Also, please mention how many times the experiment was done in each case.

Line 193 Figure 4: Once again, the purpose of this figure is not clear. It presents the same data as in Table 4. The figure does not provide any new information and actually adds to the confusion.

Line 201: “Table 4” Should be “Table 5”

Line 201 Table 5: “Percentage biofilm inhibition” I guess, you mean, “percent inhibition of biofilm formation”

Line 201 Table 5: Biofilm data as measured by OD has never been presented. Why not present that, instead of percent inhibition? Where is the control? What are these compared to? In other words, which one has 0% inhibition?

Line 201 Table 5: Concentrations that showed no growth inhibition are showing about 40% inhibition of biofilm formation? Can the authors comment on this? Why were further dilutions not tested to determine the minimum concentration that can inhibit biofilm formation?

Line 211: “Figure 5. Relationship between biofilm inhibition and AHL production” The figure shows neither biofilm inhibition nor AHL production but shows the diameter of the zone of inhibition. So, justification of the title is not clear and again, the purpose of this figure is not clear. 

Line 211 Figure 5: Are the data in this figure statistically significant enough to be presented in a paper? In majority of the results, the error bar is greater than the average. There is too much variation in the whole experiment, which brings to question the validity of the method for the experiment.

Line 208: “Where….” This sentence cannot start with “Where”

Line 208: “represent the extract doses” Specify how much these doses are.

Line 203-210: It is not clear what is being discussed here and what the figure is showing.

Line 228: “separated bacteria” Not clear what this means. Do you mean, “clinical isolate”?

Line 229: “bacteria were more energetic and active” How is that determined?

Line 234: “concentration of antibacterial drugs frequently affects their effectiveness”. Is that not obvious? Why does the obvious fact need a special mention and also have an associated reference?

Line 248: “environment from which they are separated” I don’t understand use of the word “separated”. Do you mean “isolated”?

Line 214-284: The Discussion section is too long and repetitive. It can be written more succinctly.

Some comments on quality of English along with the main comments above.

Author Response

dear reviewer,

send you an updated revised manuscript 

best regard

dr. salah Mahdi Ibrahim

Reviewer 2 Report

The manuscript has been reviewed now and the author has done Metronidazole Potentiation by Panax Ginseng and Symphid- 2 ium Officinale: A New Strategy for P. gingivalis Infection Con- 3 trol The following points need to be addressed for consideration of the manuscript:

Line 40 The microorganism name showed to be in italics

kindly do the changes where ever required in the whole manuscript

The manuscript should be improved in terms of the whole scientific approach 

The author has taken a clinical isolate of P. gingivalis however ethical approval is not mentioned in the manuscript as its necessary to take approval from the ethical committee while working with any clinical specimen. The author needs to mention the details in the material and methods.

The method for preparation of the extract should be included in details in the material and methods

results and discussion part are not written well. 

The result and discussion need to be improved. The mechanism for biological activities  (antibiofilm, anti-quorum sensing, and antimicrobial activity). The author did a synergistic study utilizing antibiotic and plant extract but didn't discuss the mechanism of the mode of action and how these are responsible for the inhibition of biological activities.

Grammatical errors are there that should be rectified. 

the author should add a characterization part for the identification of potential bioactive present in the herbal extract taken for the study. 

The manuscript has been reviewed now and the author has done Metronidazole Potentiation by Panax Ginseng and Symphid- 2 ium Officinale: A New Strategy for P. gingivalis Infection Con- 3 trol The following points need to be addressed for consideration of the manuscript:

Line 40 The microorganism name showed to be in italics

kindly do the changes where ever required in the whole manuscript

The manuscript should be improved in terms of the whole scientific approach 

The author has taken a clinical isolate of P. gingivalis however ethical approval is not mentioned in the manuscript as its necessary to take approval from the ethical committee while working with any clinical specimen. The author needs to mention the details in the material and methods.

The method for preparation of the extract should be included in details in the material and methods

results and discussion part are not written well. 

The result and discussion need to be improved. The mechanism for biological activities  (antibiofilm, anti-quorum sensing, and antimicrobial activity). The author did a synergistic study utilizing antibiotic and plant extract but didn't discuss the mechanism of the mode of action and how these are responsible for the inhibition of biological activities.

Grammatical errors are there that should be rectified. 

the author should add a characterization part for the identification of potential bioactive present in the herbal extract taken for the study. 

Author Response

dear reviewer2,

send you an updated revised manuscript 

best regard

dr. salah Mahdi Ibrahim

Round 2

Reviewer 1 Report

Metronidazole Potentiation by Panax Ginseng and Symphidium Officinale: A New Strategy for P. gingivalis Infection Control 

by Ibrahim et al. 

The authors have addressed all my concerns. The manuscript is now of acceptable quality.  I have two minor comments:

Line 369: “insolationan” I guess, the authors mean “isolation”

Line 370: Add spaced after “producing”

Line 369-370: This reference is not searchable in Pubmed, so I could not access the paper. From the title, it does not appear to be on mechanism of action of metronidazole. Of all the pioneering work on mechanism of action of metronidazole, the authors listed a paper that is not listed in Pubmed. 

Line 414: Delete “.” At the start of the line. 

Author Response

dear reviewer 1

I send the response 2

best regard 

dr.salah Mahdi Ibrahim

Reviewer 2 Report

The author made all the corrections as mentioned in the comments hence can be accepted for publication

Author Response

dear reviewer 2

I send the response 2

best regard 

dr.salah Mahdi Ibrahim
